# High-Performance Polyurethane Nanocomposite Membranes Containing Cellulose Nanocrystals for Protein Separation

**DOI:** 10.3390/polym14040831

**Published:** 2022-02-21

**Authors:** Víctor-Hugo Antolín-Cerón, Francisco-Jesús González-López, Pablo Daniel Astudillo-Sánchez, Karla-Alejandra Barrera-Rivera, Antonio Martínez-Richa

**Affiliations:** 1Departamento de Ciencias Básicas Aplicadas, Universidad de Guadalajara, Tonalá 45425, Mexico; vhaceron@hotmail.com (V.-H.A.-C.); frankii9721@gmail.com (F.-J.G.-L.); pablo.astudillo@cutonala.udg.mx (P.D.A.-S.); 2Departamento de Química, Universidad de Guanajuato, Guanajuato 36050, Mexico; fionita@ugto.mx

**Keywords:** microfiltration membrane, nanocellulose, nanocomposite, albumin rejection, polyurethane

## Abstract

With the aim of exploring new materials and properties, we report the synthesis of a thermoplastic chain extended polyurethane membrane, with superior strength and toughness, obtained by incorporating two different concentrations of reactive cellulose nanocrystals (CNC) for potential use in kidney dialysis. Membrane nanocomposites were prepared by the phase inversion method and their structure and properties were determined. These materials were prepared from a polyurethane (PU) yielded from poly(1,4 butylene adipate) as a soft segment diol, isophorone diisocyanate (IPDI) and hexamethylenediamine (HMDA) as isocyanate and chain extender, respectively (hard segment), filled with 1 or 2% *w*/*w* CNC. Membrane preparation was made by the phase inversion method using *N,N*-dimethylformamide as solvent and water as nonsolvent, and subjected to dead-end microfiltration. Membranes were evaluated by their pure water flux, water content, hydraulic resistance and protein rejection. Polymers and nanocomposites were characterized by scanning electronic and optical microscopy, differential scanning calorimetry, infrared spectroscopy, strain stress testing and ^13^C solid state nuclear magnetic resonance. The most remarkable effects observed by the addition of CNCs are (i) a substantial increment in Young’s modulus to twenty-two times compared with the neat PU and (ii) a marked increase in pure water flux up to sixty times, for sample containing 1% (*w/w*) of CNC. We found that nanofiller has a strong affinity to soft segment diol, which crystallizes in the presence of CNCs, developing both superior mechanical and pure water flow properties, compared to neat PU. The presence of nanofiller also modifies PU intermolecular interactions and consequently the nature of membrane pores.

## 1. Introduction

Polyurethane (PU) has been widely applied in many industries as a versatile material due to its relatively easy manufacture, low cost, the accessible source of their raw materials and because they are typically used as glues, fibers, coatings and hoses among other applications. Recently, PUs have been used in medical and environmental applications such as in the removal of dyes and organic solvents and as filters for macromolecules [1]. Many researchers have used PU for the preparation of membranes because of its high flux capability, high salt rejection properties and high hydrophilicity [2,3]. However, PU lacks enough good mechanical and thermal properties by itself, and porosity is in general not adequate for certain applications in membrane science. 

Polymer nanocomposites reinforced with a low fraction of nanofillers have received great attention due to the fascinating properties that they present, and could compete with those of the most advanced materials available in the market [4,5]. Cellulose nanocrystals (CNCs) are highly crystalline rod-like nanomaterials isolated from cellulose fibers by acid hydrolysis [6]. CNCs have been incorporated into a wide variety of polymer matrices as reinforcing fillers, due to their intrinsic properties such as nanoscale dimensions, high surface area, unique morphology, low density, high specific strength and Young’s modulus, as well as a very low coefficient of thermal expansion [7,8,9].

Some of the most important characteristics that dialysis membranes must have are: (i) good biocompatibility; (ii) affordable cost; and (iii) appropriate morphology, from which PUs are one important choice. Among the materials with these characteristics, polyurethanes (PU) stand out as good candidates for these applications. Another particular feature of these materials is the hydrophilicity enhancement when they are reinforced with CNCs, allowing also for the elimination of macrovoids during membrane preparation [10,11]. Additionally, Bai et al. [12] reported that the poly(vinylidene fluoride) (PVDF) membrane containing low amounts of CNC (from 0 to 0.25 wt%) shows increased water flux, mechanical strength and thermostability. Due to their singular hydrophilicity, CNCs are considered as attractive to reinforce membranes, as they improve water permeability and the capability to remove organic substances, metal ions, dyes and microbes [11]. The wettability of membranes can then be improved and their biocompatibility enhanced by adding CNC, instead of using other approaches such as the chemical modification of polymer. This procedure also increased the diffusive transport properties of solute through the membrane. CNC has been proven to act as a pore forming agent, which is important in attaining appreciable porosity and pore size of the membrane [13].

The molecular weights of proteins, which are believed to cause various chronic side reactions in dialysis patients, are in a range of 10–55 kDa. In particular, it is well known that the accumulation of 2-microglobulin (2-MG, 11,800 Da) in the body causes amyloidosis [14]. In hemodialysis separation, low- and middle-molecular-weight uremic toxins such as urea, uric acid, creatinin and 2-microglobulin have to be removed from blood. However, proteins such as serum albumin (66 kDa) should be retained. Therefore, the desirable kidney dialysis membranes must have a suitable pore size distribution which can selectively act for the blood components previously mentioned. Additionally, higher mechanical strength can be reached when the membranes possess a sponge-like asymmetric structure bearing high transmembrane pressures and preventing membrane rupturing and leakage, which might happen in the membrane structure with macrovoids and after continuous use [13].

In a previous work, we reported the synthesis and characterization of a biodegradable membranes filled with functionalized carbon nanotubes and showed evidence that nanofiller provides sites for hydrogen bonding interactions in the matrix which influence the mechanical properties and the porous morphology of the membrane [15]. In this work, filtration of egg albumin was attempted in order to explore the properties of and interaction between PU and CNC for hemodialysis membranes. The optimal experimental conditions to obtain a high-performance hemodialysis membrane for protein separation are reported. 

## 2. Materials and Methods

### 2.1. Materials

Poly(1,4 butylene adipate) (Diexter G 4400-57) with an hydroxyl equivalent weight of 984 was supplied by Coim USA Inc. (West Deptford, NJ, USA), isophorone diisocyanate (IPDI) from Watane. Hexamethylene diamine, Tin (II) 2-ethyl hexanoate and *N,N* dimethyl formamide (DMF) were acquired from Aldrich (Toluca, México). Dichloroethane and absolute ethanol were purchased from Merck (Naucalpan, México), ethylenediamine from Spectrum (Jersey City, NJ, USA), NaH_2_SO_4_ from Honeywell Fluka (Charlotte, NC, USA), chloroform 99.8% from Karal México, and Na_2_HSO_4_ and egg albumin from Meyer (Mexico City, México). All reagents were dried before being used.

### 2.2. Synthesis of Cellulose Nanocrystal

CNCs were prepared by acid hydrolysis of commercial cellulose microcrystals (MCCs), following a method reported in the literature [16,17,18]. First, 20 g of MCC and 175 mL of sulphuric acid solution (64% (*w/w*)) were mixed in a 250 mL three-neck round-bottom flask and homogenized with a mechanical stirrer. Hydrolysis was carried out at 45 °C for 30 min. The obtained product was diluted in 4 L of deionized water to stop the hydrolysis reaction. Next, to remove the acid excess, the suspension was centrifuged and 1 L of the CNC suspension was obtained. Suspension was dialyzed for 5 days for neutralization. To purify the suspension, ion exchange resin (Dowex Marathon MR-3 hydrogen and hydroxide form) was added and stirred for 24 h and then removed by filtration. The pH of the CNC suspension was adjusted to around 9.0 adding dropwise upon stirring 1.0% NaOH aqueous solution [17,18,19]. The CNC suspension was sonicated in order to obtain a stable suspension of the nanocrystals, which was stored in a fridge at 3 °C to avoid bacterial growth.

### 2.3. Synthesis of PU and Their Nanocomposites

Dry polyester diol and isophorone diisocyanate (OH:NCO mol ratio 1:2) (2.5 g:0.57 g, 2.58 meq:5.16 meq) were charged into a round-bottom flask in which a solution of 1,2 dichloroethane (20 mL) containing stannous 2-ethyl hexanoate (18 μL, 0.055 mmol) was previously placed. The mixture was stirred for 3 h at 80 °C and then the chain extender (HMDA, 0.15 g, 2.58 meq) was added dropwise. The reaction mixture was kept at the same temperature for 3 h. For nanocomposite synthesis (see Figure 1), the specific amount of CNC was added to the initial solution of polyester diol and IPDI in 1,2 dichloroethane, and this mixture was sonicated for 15 min. Then the catalyst was added, and the reaction mixture was heated at 80 °C under stirring for 3 h. HMDA was added dropwise and the reaction was maintained for 3 h at 80 °C. The product was poured over a leveled aluminum mold at room temperature for 24 h. Evaporation of the solvent was carried out until a film was obtained. The film was released and dried in vacuum at room temperature for 12 h. 

### 2.4. Membrane Preparation 

Membranes were prepared following the phase inversion procedure reported in the literature [20,21,22]. A 20% wt polymer or nanocomposite solution using *N,N*-dimethylformamide was prepared. The solution was casted on a Petri dish until a thin layer was formed at room temperature. A distilled water container was used separately as a gelation bath at room temperature. The petri dish was immersed in the bath and kept underwater during the night in order to ensure efficient mass transfer. The formed membrane was washed with distilled water and dried at room temperature for 48 h. The thickness of the membranes varied from 0.10 to 0.25 mm.

### 2.5. Microfiltration

An acrylic cylindrical dead-end microfiltration cell with a 1 L capacity was used to test membrane performance. The cell consists of two nylon caps and threaded rod flanges, in which an “O” ring rubber and a stainless-steel grid were inserted to hold the membrane. This microfiltration cell was connected to an air compressor with a pressure control valve and gauge through a feed reservoir.

Prepared membranes were cut into the size needed for fixing it up in the microfiltration cell of 20.2 cm^2^ area. The membranes were subjected to pure water flux studies at transmembrane pressures of 40, 98 and 198 kPa. Flux was measured under steady-state flow. The water flux was measured every 10 min. The pure water flux was determined using Equation (1) [23,24]:(1)$$J_{w}=\frac{Q}{A\;\Delta t\;}$$
where *Q* is the quantity of permeate collected (L), *J_w_* is the water flux (L m^−2^ h^−1^), Δ*t* is the sampling time (h), and *A* is the membrane area (m^2^). Measurements were carried out in triplicate.

To determine the hydraulic resistance of the membrane (*R_m_*), the pure water flux of the membranes was measured at transmembrane pressures (Δ*P*) of 49, 98 and 196 kPa. *R_m_* was evaluated from the slope of *J_w_* versus Δ*P* plot [25]:(2)$$J_{w}=\frac{\Delta P}{R_{m}}$$

### 2.6. Water Content and Porosity

The water content fraction (*ε*) and the porosity (*P*) of the membranes were calculated using Equations (3) [26] and (4) [27].
(3)$$\%\varepsilon =\frac{W_{wet}-W_{dry}}{W_{wet}}100$$
(4)$$\%P=\frac{W_{wet}-W_{dry}}{\rho Ah}100$$
where *A* is the membrane surface (m^2^), *h* is the membrane thickness (m), *ε* is the percentage of water content, *P* is the membrane porosity percentage, *W_wet_* is the wet sample weight (g), *W_dry_* is the dry sample weight (g), *ρ* is the density, *A* is the membrane area (cm^2^) and *h* is the membrane thickness (mm). 

### 2.7. Protein Rejection

After mounting the membrane in the ultrafiltration cell, the chamber was filled with protein solution and immediately pressurized to the desired level (149 kPa) and maintained at a constant level throughout the run. Albumin egg protein was dissolved (0.1%) in a phosphate buffer (0.05 M, pH 7.2) and used as standard solution. The permeate was collected over measured time intervals in graduated tubes and then analyzed for protein content by UV-Vis spectrophotometry (Hash-1000) at λ_max_ 450 nm after adding biuret reagent protein. Separation was calculated from the concentrations of feed and permeate using Equation (5) [28]:(5)$$\%S_{R}=1-\frac{C_{P}}{C_{F}}\times 100$$
where % *S_R_* is the % solute rejection, and *C_P_* and *C_F_* are concentrations of permeate and feed, respectively.

### 2.8. Differential Scanning Calorimetry (DSC) 

DSC was used to study the thermal properties of the prepared nanocomposites. The measurements were carried out on a Mettler Toledo DSC model DSC822e previously calibrated with indium. All tests were performed under a nitrogen atmosphere. Thermograms were recorded by heating the samples from −90 to 80 °C at 10 °C/min; only the second scan was reported. Sample weights ranged between 5 and 10 mg. The glass transition temperature (T_g_) of the polymer matrix was evaluated by the inflexion point criteria and the melting enthalpy was calculated from the area under the endothermic peak.

### 2.9. Mechanical Test

Tensile stress–strain tests were performed at a deformation rate of 50 mm/min in a United machine model SFM-10. The tested samples had a rectangular prism shape (55 × 12 × 0.5 mm) according to the ASTM D882 standard test method. Five samples were tested for each polymer composition.

### 2.10. Infrared Spectroscopy (FTIR)

FTIR spectra of the CNC, PU and nanocomposites were obtained on an Alpha II spectrophotometer from Bruker, using an attenuated total reflection (ATR) unit. Each spectrum is an average of four scans with a resolution of 2 cm^−1^.

### 2.11. Optical Microscopy

A homemade dual slide test-cell configuration was used to explore two-dimension morphology. A spacer formed by three coverslips packed was placed between the two slides and the assembly was held together with a commercial plastic clamp. This device was submerged into a coagulation bath in order to isolate the casting solution between the two slides. The microscopy visualization studies of cover morphology and size porous developed during the dry-cast process of the PU/DMF/H_2_O system were performed using an Amscope optical microscope. Images were recorded with the aid of a cell phone camera.

### 2.12. Scanning Electronic Microscopy (SEM)

A cross-section morphology of the membranes was examined with a field emission scanning electron microscope (FE-SEM), model MIRA 3LU of Tescan (Brno, Czech Republic) with an accelerating voltage of 15 kV. Samples were frozen in liquid nitrogen, broken, mounted into the sample holder and gilded for 20 s. 

### 2.13. Water Contact Angle (WCA) Measurement

For the water contact angle measurement, a drop of water was placed on the surface of the membrane film with a pipette, and the contact angle between the water drop and film was measured.

### 2.14. Carbon-13 Solid-State NMR

Solid-state carbon-13 NMR spectra were recorded under proton decoupling on a Bruker Avance 400 operating at 100.613 MHz for ^13^C. A Bruker probe equipped with 4 mm rotors was used. CP-MAS spectra were obtained under Hartmann–Hahn matching conditions and a spinning rate of 6.0 kHz. A contact time of 2.5 ms and a repetition time of 4 s were used. The measurements were made using spin-lock power in radiofrequency units of 60 kHz and typically 4000 transients were recorded. Chemical shifts were externally referenced to tetramethylsilane using adamantane.

## 3. Results

Table 1 lists the composition for each sample analyzed in this work. As is well recognized in the field of cellulose nanocrystal-based nanocomposites, in order to obtain a better performance it is essential to achieve a better dispersion and hence a better polymer matrix-CNC interaction [29]. Based on our findings in a previous publication [15], two nanocomposite compositions (1 and 2 % wt) were studied. 

### 3.1. FTIR Characterization

Figure 1a shows the characteristic broad peak in the 3500–3200 cm^−1^ region for free O–H stretching vibration for CNC. C–H stretching vibration around 2894 cm^−1^ is also present [30]. Peaks in the region 1649–1641 cm^−1^ are attributed to the O–H bending of the adsorbed water [31]. The peak observed at 1054 cm^−1^ is due to the C–O–C pyranose ring (antisymmetric in phase ring) stretching vibration. The band at 902 cm^−1^ is associated with the β-glycosidic linkages between glucose units in cellulose nanocrystal [32]. A C–C ring breathing mode is located around 1155 cm^−1^, whereas a C–O–C glycosidic ether band in CNC appears at 1105 cm^−1^ [33]. 

Figure 1b shows an expansion zone of carbonyl of the FTIR spectra of PU and nanocomposites. The ester carbonyl band of the pure PU is centered at around 1720 cm^−1^. A shoulder around 1640 cm^−1^ (inset Figure 1b), due to the stretching of the urea group, is present. This band is attributable to “ordered hydrogen bonds” among urea groups [34]. The decrease in intensity for the urea band in the nanocomposite spectra suggests that intermolecular interactions decrease in nanocomposites when CNC is added. This effect is related to the formation of hydrogen bonding between the urea groups and hydroxyl groups of CNC. This fact indicates that additional hydrogen bonds between suitable chemical groups of CNC and carbonyl groups of hard and soft segments of PU matrix were created. The decreased intensity in the urea band at 1640 cm^−1^ by hydrogen bonding formation between PU segments and chemical groups of CNCs suggests that the former peak is related to not very well-defined nanocomposite species. This observation suggests that it is very likely the occurrence of hydrogen bonds between PU-CNC, and simultaneously the hydrogen bond breaking of the urea-moiety-ordered bonds. This leads to a decrease in the viscosity of cast solution nanocomposites. Formation of hydrogen-bonded species can be also associated with a better mass transfer during the phase inversion process. 

In Figure 1c, the stretching region between 3100 and 3600 cm^−1^ is depicted. A band around 3340 cm^−1^ is observed in PU and nanocomposites attributed to hydrogen-bonded N-H groups, whereas CNC shows two peaks at 3338 and 3268 cm^−1^ ascribed to OH group stretching vibration [35]. In NC 2, an increase in the peak intensity of the OH stretching vibration band occurs as the amount of CNC content is increased. A lower occurrence of hydrogen bonds in nanocomposites is observed when CNC content is below 2 wt%, which leads to a good dispersion of nanocomposites in the solvent. The hydrogen bonds are disrupted due to the presence of CNC; as a consequence, the dispersion shows shear thinning [36].

### 3.2. DSC Measurements

Figure 2 shows the thermograms obtained for samples by DSC. In Table 2, data obtained from the DSC experiments are displayed. Glass transition temperatures remain unchanged at −53 °C in all PUs. In particular, endotherm peaks at 45 and 43 °C are detected for NC 1 and NC 2, respectively; these peaks are ascribed to the melting of the soft segment domains. In the amorphous region, the fact that glass transition temperatures remain constant in all samples indicates that the addition of CNC to PU matrix has no significant effect on the motion of SS. The observation of a melting endotherm in both nanocomposites indicates that nucleation of polymer crystals is promoted by the presence of the well-dispersed nanofiller. As Habibi et al. stated [37], a defined melting endotherm appears as the polyester diol moiety crystallized on the cellulose nanocrystal surface. On the other hand, a smaller crystalline peak is present in the NC 2 thermogram. Low crystallinity was expected as the chain ordering for crystallization is impeded due to the high concentration of CNC, suggesting that the crystalline structure of the CNC limited the crystallization of the polyester diol moiety. Thus, the attached diol chains on the surface CNC of NC 2 are more likely to exist in non-crystalline domains [38,39]. 

### 3.3. Stress–Strain Testing

Figure 3 shows the stress–strain curves of pure PU and its nanocomposites prepared with 1 and 2 wt% of CNC. Results are summarized in Table 3. In stress–strain curves of the PU and its nanocomposites, three distinct zones can be distinguished. (1) First is a linear elastic zone in which the inset shows the slope from which Young’s modulus can be calculated. This initial section of the curve is governed by the crystallization of soft segment (SS) and local reordering of the PU macromolecules. Strain–stress curves indicate that CNC preferentially reinforces the soft segment microdomains, instead of the PU hard segments. Moreover, nanocomposites present higher stress values than neat PU, and the appearance of the yield point observed in nanocomposites could be due to the fact that CNC creates a nucleating point in SS since CNC is a crystalline material. (2) Second is a second zone, which extends until a maximum stress is reached. This is related to the breaking-up of the interconnection network of hard segments. (3) The third zone is that of plastic deformation, which corresponds to strain-induced soft segment crystallization as well as the further breaking of the hard microdomains [40]. The behavior observed in the third zone suggests a decrement in the elongation properties of the material, which could be related to hydrogen bond formation between the hydroxyl groups of the CNC and PU in both segments, which in turn inhibits the association between the chemical groups of the PU matrix [15]. A concentration of 1 % wt CNC content seems to decrease this association even more, but the effect is more pronounced as a consequence of a better dispersion of reinforcement. In addition, the higher values of stress data indicate that CNCs orient strongly at high strains and also induce synergistic PU orientation effects, contributing to an increase in strength level after this stage. 

The introduction of CNCs led to an improvement of the Young’s modulus and tensile strength in both nanocomposites. After adding 1% of CNC, the tensile strength and elastic modulus increased more than 4 times and 22 times, respectively, reaching a value of 9.3 Mpa for tensile strength and 220 Mpa for elastic modulus. However, samples of NC 2 showed lower performance in mechanical properties in comparison with NC 1. Change of tensile strength and elastic modulus coincides with other results from other authors: CNCs for the studied sample behave as a point of failure in the composite, and probably interfere in the formation of the PU network, as hydroxyl -OH groups in cellulose can also react with the isocyanate group [41,42,43,44,45]. Taking into account the reinforcement L/D aspect ratio, the theoretical volume fraction of CNC needed in order to reach the percolation threshold in the nanocomposite was estimated by means of the following equation [8]:(6)%VRC=0.7LD×100

In a previous report from our research group, the average length of CNC (determined by image analysis), was found to be between 100 to 300 nm in length and between 5 and 10 nm in width when 1–2% of reinforcement is used [17,18,19]. According to our results, it is very likely that when 1 wt% of CNC is used, percolation of the system is not reached, and a network of dispersed crystals interpenetrates the PU matrix, leading to improved mechanical properties. Higher concentrations of the crystals result in agglomeration, and thus the properties drop. Since the sample is mainly heterogeneous and agglomerates, stress concentrators are created and there are more points of failure within the material [46,47,48]. Decreasing in elongation at break can be attributed to the restriction of polymer segments, which is caused by the existence of nano- and microphase separation between the rigid CNCs nano-filler domains and the PU matrix.

### 3.4. Carbon-13 Solid-State NMR

Figure 4a shows the ^13^C CP-MAS NMR spectrum of neat PU and their nanocomposites. An intense peak centered at 174 ppm is ascribed to carboxyl carbon from a polyester diol moiety. A broad peak centered at 159 ppm and a shoulder at 157 ppm are observed in the carbonyl region and are ascribed to the urea and urethane carbonyls, respectively. The peak at 65 ppm is ascribed to those soft-segment diol carbons that are adjacent to oxygen, while the small shoulder around 45 ppm is ascribed to the hard-segment carbons that are adjacent to a urea moiety. The peak at 33 ppm is associated with the primary carbons in the cyclohexyl rings. It is interesting to note that the line width of the 174, 65, 35 and 25 ppm peaks of NC 1 is remarkably sharp as compared with other peak samples (see Figure 4b,c). This feature implies that an effective motional narrowing occurs, possibly due to the segmental motion of the polymer chain and the existence of ordered arrays. This narrowing indicates that the presence of CNC causes a narrower distribution of the soft-segment environments and/or a decrease in the segmental motion [49,50,51,52]. The latter issue results from the interaction between the CNC and the soft segments. 

Narrow and displaced peaks are detected for samples with low contents of CNC. In the inset of Figure 4c, this feature is observed for NC 1, and peak shifts upfield with respect to PU signals. As CNC content increases, peaks become broader and also shift upfield. In that regard, it is observed that the peak shifts upfield from 64.6 (PU) to 64.4 and 64.3 ppm for NC 2 and NC 1, respectively. These results indicate that the soft-segment chain conformations change drastically due to the presence of CNCs. Conformational changes of the soft-segment chain for samples with 1% of CNC lead to strong differences in physical properties. It is also found that the mass fraction of the crystalline component agrees well with the presence of crystallinity determined from the enthalpic endotherm observed in the thermogram curve shown in Figure 2. 

### 3.5. Microfiltration 

Microfiltration membranes based on PU and cellulose nanocrystals were prepared with 1 and 2% *w/w*. When the polymer solution was immersed in a gelation bath, a change in appearance took place from a transparent to a spongy-like white color material, indicating that mass transfer had taken place [53].

Figure 5a–c shows the curves obtained at different transmembrane pressures. The membranes were subjected to a 40, 98 and 196 kPa, under steady-state conditions and at a constant sampling period. Three runs were carried out for each sample, and the recorded average values are reported in Table 4.

For all membranes, the steady-state flux was obtained within 2–3 h, allowing for recording pure water flux properly. PU and nanocomposites show a linear increase of pure water flux with respect to the pressure transmembrane. This behavior is due to the fact that during the filtration, the walls of the pores become closer and denser, which leads to the reduction of the pore size and consequently a uniform flux [54]. The values recorded from the curves of Figure 5 indicate that nanocomposites show an important increase in the flux compared with neat PU membrane. The role of the CNC on final membrane morphology is related to the fact that CNC has a hydrophilic nature [55]. When casted solution of nanocomposites is submerged into the coagulation bath, the mass transfer is more effective compared to the pure PU casted solution. This fact can cause the mixture to reach the unstable region rapidly, resulting in a spinodal decomposition. Stratham et al. stated that formation of macrovoids is the result of the rapid penetration of nonsolvent at certain weak spots created by CNC in the membrane [56]. Macrovoids are also usually associated with an increase in miscibility. The tendency toward macrovoid formation will increase with a lower content of CNC probably because of the well-dispersed nanofiller and the more likely formation of covalent bonds between PU and CNC. At higher concentrations of CNC, the reactive hydroxyl groups promote the formation of a rigid network through hydrogen bonding of nanofiller that is governed by the percolation mechanism. For 2 % wt CNC concentration, the network is expected to form above the critical volume fraction at the percolation threshold. Barzin et al. reported that a working flux of pure water for blood hemodialysis membranes is in the range of 30 to 280 L m^−2^ h^−1^, suggesting that prepared membranes are suitable for kidney dialysis at different transmembrane pressure [13]. 

The high flux observed in nanocomposites is due to the formation of channeling morphology by the succession of interconnected pore structure formed in the polymer poor phase by spinodal decomposition, which provides the route for easy solvent penetration. The dominant morphology on membrane nanocomposites is formed from spinodal decomposition. Once phase separation occurs via spinodal decomposition, a higher flux of water is expected due to the continuous formation of the polymer-poor phase [57]. 

Another important effect on porosity is the fact that the cellulose nanocrystals have active groups on the surface of CNCs that can form hydrogen bonds with the coagulation bath (water) [58], which maximizes the mass transfer process during phase inversion.

Figure 5 shows the three lines corresponding to the hydraulic resistance of prepared membranes in which the low slope values indicate higher hydraulic resistance on water; thus, the PU membrane reveals a higher resistance flux, in contrast with pure water flux decreasing in both nanocomposites. For nanocomposite samples 1 and 2, increments were 9 and 18 times higher with respect to neat PU, respectively. 

The effect of the concentration of CNCs (i.e., 1 and 2 wt%) in membrane performance at different transmembrane pressures—40, 98 and 196 kPa—are shown in Figure 5. In Table 5 the slopes of the linear relation of flux to applied pressure are listed, which can be directly related to the transport resistance to water [59]. It is evident from the figures that an increase in transmembrane pressure increases the flux at a linear rate. In NC 1, the increase in flux is greater than those observed for neat PU and NC 2. Resistance for neat PU and NC 2 is higher, resulting in lowered flux of pure water and lower membrane resistance (Rm). 

### 3.6. Photomicrography 

To assess the effect of CNC content on the hydrophilicity of membranes, the water contact angles (WCA) of neat PU and its nanocomposites were measured. As shown in the upper part of Figure 6, the WCA was 118, 68 and 73° for neat PU, NC 1 and NC2, respectively. The WCA of nanocomposites was smaller than that of the neat PU, which confirms the increase in hydrophilicity on membranes’ top surface by the addition of the CNC. 

Figure 6 also shows the micrographs of the top surface membrane morphology of the PU and its nanocomposites, and reveals the existence of two different morphologies. (i) Figure 6a reveals uniform-shape porous materials formed from the nucleation and growing mechanism of neat PU during mass transfer, which induces the surrounding polymer-rich phase to form a well-defined cellular structure. (ii) Figure 6b,c presents a different structure formation, formed from a spinodal decomposition mechanism of both nanocomposite solutions, which is characterized by (a) interconnection of phases in a bicontinuous network and (b) irregular and random porous distribution. We can attribute the presence of cavities in Figure 6 to macrovoids formed on the top layer surface [22].

The SEM micrographs for the cross-sections of neat PU and its nanocomposite membranes are shown in Figure 7. The samples were frozen in liquid nitrogen and broken into their collapsed form. SEM images are alike in morphology on the top-surface micrographs, since both show the same porous pattern. An analysis of the micrographs indicates that the highest diameter for macrovoids was 39 µm for NC 1, whereas for neat PU a value of 30 µm was recorded. The pore size measurement in membrane cross-sections resulted in 1.6 ± 0.1 µm for both neat PU and NC1. This last observation suggests that the prepared membranes can be used for microfiltration, as the pore size is greater than those seen for ultrafiltration applications, typically 0.001–0.5 µm, at the middle zone in the membrane [60].

Recorded values for water content (%), porosity (%) and S_R_ (%) for samples are reported in Table 6. Porosity and water content in the PU membrane were found to be 52 and 74%, respectively. Two novel features are distinguished when CNC is incorporated. (i) First, the porosity increases in both nanocomposites, confirming that the hydrophilicity of CNC promotes mass transfer and absorption of more water during the inversion phase, creating larger cavities in membrane nanocomposites. (ii) Conversely, the water content of the nanocomposites decreases due to the co-continuous porous morphology and makes more difficult the retention and absorption of water. At higher concentrations, the CNC acted as a crosslinking agent in a polymer matrix, restraining the swelling mechanically. This phenomenon leads to lower values for water content and porosity. The protein rejection reduces in the presence of 1 % wt of CNC. Similar values were recorded for neat PU and 2% CNC nanocomposite. These results suggest that larger proteins in the bloodstream, such as serum albumin, can be retained (66.5 kDa) by these membranes, whereas smaller proteins in dialysis filtration (in the range of 10–55 kDa) can be removed. Egg albumin (43–46 kDa) is a good model for establishing the scope of membrane retention based on protein size, and the results indicate that the membranes obtained are useful for separation in the range required for kidney dialysis applications [61,62].

## 4. Conclusions

Polyurethane nanocomposite membranes were effectively synthesized and tested for aqueous separation of egg albumin protein by microfiltration. CNC plays an important role in controlling pore size and flux. We find that nanocomposite pore formation occurs via a spinodal decomposition mechanism, whereas for neat PU the nucleation and growth mechanism operates. The formation of a co-continuous structure is attributed to the growing of the polymer-poor phase by coalescence after spinodal decomposition, as well as the flux of nonsolvent into the polymer-poor phase due to the presence of dispersed CNC. There is enough evidence that the CNCs are hosted in a SS, as evidenced by: (i) the increase of Young’s Modulus (governed by crystals of diol) in the linear zone of the strain–stress curve, (ii) the melting endotherms recorded by DSC, and (iii) narrower peaks observed for carbons of polyester diol moiety observed in the Carbon-13 CP-MAS NMR spectra of NC 1 sample. Results of tensile strength and elastic modulus showed that 1 wt% of CNC is the optimal concentration to improve the mechanical properties of PU. Improvement in mechanical properties is indeed significant by the presence of the CNC. Pure water flux had a wide range of values, which provides the versatility needed for uses in biomedicine. Protein separation on the prepared materials is suitable for kidney dialysis, since NC 2 and neat PU can remove proteins under the molecular size of the serum albumin protein. NC 1 retains less protein than the other membranes. This behavior is mainly attributed to (i) a decrease in the degree of phase separation between hard and soft segments and (ii) intermolecular interactions (bidentate hydrogen bonds) among urea groups, as detected by infrared spectroscopy. These facts lead to a change in the shear thinning of polymer solution and also to the hydrophilicity of CNC, which supports the mass transfer in the phase inversion process, leading to higher porosity in the membrane.

## Data Availability

Data presented in this study are available on request from the corresponding author.

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
