# Peer review of "High-Performance Polyurethane Nanocomposite Membranes Containing Cellulose Nanocrystals for Protein Separation"

_polymers, 2022, doi:10.3390/polym14040831_

Round 1

Reviewer 1 Report

This paper details the preparation of PU and PU doped with 1% and 2% CNC membranes as well as the chemical, physicochemical and mechanical characterization using solid state NMR, IR, DSC, optical microscopy, SEM and tensile stress-strain tests. Furthermore, the hydraulic resistance, the porosity, the water content and the protein rejection were measured.

I think it is desirable to reconsider it after addressing the following minor and major issues.

  • The scheme 1 needs an improvement. It is appeared that the CNC takes place in the reaction.
  • In Figure 1, every sample needs a different line shape of color.
  • In Figure 3, what are the axes in the inside diagram?
  • In Figure 4, the same with the figure 1.
  • Better explanation of figure 6. I don’t think that the hypothesis in the lines 466-474 can be concluded by the optical microscopy.
  • In figure 7, Do these SEM photos correspond to collapsed or to swelled membranes? I thing that SEM images from the swelled membranes will give better insights. You can perform SEM in freeze-dried membranes in order to detect the real pores.
  • The protein rejection part needs improvement. I think this part is the main reason of this work. The authors studied the protein rejection of egg albumin which has molecular weight 43-46 kDa. I think that the authors have to studied the rejection of proteins with molecular weight lower than 10 kDa and higher than 55 kDa in order the membrane to be capable for kidney dialysis application.

Author Response

Reviewer 1

1) CNC structure was eliminated from the Scheme 1, in order to avoid confusion in the role that reinforcement plays in the nanocomposite

2) Figure 1 was amended accordingly, showing a different color for each curve

3) In this new version, axes are labeled in figure 3 inset as requested

4) In this new version, a different color for each curve is used in Figure 4

5) Optical microscopy is a versatile technique for obtaining membrane formation images. The interpretation of optical microscopy images to evidence the morphology changes during wet phase inversion process has been reported in the literature (see Melnig, V.; Apostu, M. O.; Tura, V.; Ciobanu, C.; Optimization of polyurethane membranes: Morphology and structure studies, Journal of Membrane Science, (2005) 267(1-2) 58-67). We based our discussion in this reference.  

6) The SEM photos in Figure 7 correspond to collapsed membranes. This is now specified in the manuscript (see rows 478-479). The pore measurement was omitted.

7) Studies on the protein rejection of albumin proteins (from 20 to 65 kDa range) has been reported by other authors using similar membrane materials.  There are subtle differences in the protein rejection process for  these proteins (see Sivakumar, M.; Malaisamy, R.; Sajitha, C. J.; Mohan, V.; Rangarajan, R.; Preparation and performance of cellulose acetate-polyurethane blend membranes and their applications-II. Journal of Membrane Science, (2000) 169(2) 215-228). In this work we test egg albumin knowing that this protein is in the range of proteins reported, and taking into consideration that most of plasma proteins are far above in molecular weight, compared to blood protein albumins.

We are now working with higher and lower molecular weight albumins, but we are not able to conclude the experiments on the due date required by the Journal.

Reviewer 2 Report

  1. What is the difference between NC1 and NC2? The authors need to mention it at a prominent place in the text.
  2. The DSC measurement needs to be repeated with the starting temperature of at least of -50 C, because the glass transition region of the samples is not so obvious as it should be.
  3. Why no endotherm peak is found for NC2 sample in the DSC curve? However, why there is a melting temperature for NC2 in Table 2?
  4. How about the uniformity of the thickness for the samples? The stress-strain curves in Figure 3 are so discontinuous and unsmooth, and the authors need to check their samples and repeat this experiment if possible. Please check the unit or the value of strain in Figure 3.

Author Response

Reviewer 2

1) The characteristics of samples NC 1 and NC 2 are indicated in table 1. They are related to the concentrations of CNC used in this work. A brief discussion on the amounts used for reinforcement is now included at the beginning of the Results and Discussion section

2) DSC measurements were now obtained at the requested temperature range, in order to obtain a more precise value of glass transition temperature Tg .

3) In the new DSC thermogram, endotherms and the melting temperatures for all samples (and in particular for NC2 sample) are now better observed

4) Stress-strain curves obtained from all the samples in the mechanical tests were smoothed using a curve smoothing algorithms and are shown in a new Figure 3. The observed noise is the result of the low sensitivity of the loaded cell using in the measurements.

Round 2

Reviewer 1 Report

The authors addressed the requested changes. So, i recommend to publish this manuscript in the present form.